# Complications following transcutaneous cecal trocarization in horses with a cattle trocar and a cecal needle

**Renata Gebara Sampaio Dória** *[○], **Gustavo Morandini Reginato**[○], **Yumi de Barcelos Hayasaka**[○], **Paulo Fantinato Neto** [○], **Danielle Passarelli**[○], **Julia de Assis Arantes**[○]

Faculty of Animal Science and Food Engineering, Department of Veterinary Medicine, University of São Paulo (USP), Pirassununga, São Paulo, Brazil

○ These authors contributed equally to this work.
* redoria@usp.br

**Data Availability Statement:** All relevant data are within the paper.

**Funding:** The authors thank São Paulo Research Foundation (FAPESP), grant numbers 2012/00632-

## Abstract

Percutaneous decompression of the cecum is a procedure that could be considered for horses with cecal gas distension. The aim of this study was to identify complications such as peritonitis and clinically relevant peritonitis (CRP) after transabdominal cecal trocarization in healthy horses using a cattle trocar and a cecal needle. Mixed breed horses were assigned to three groups (n = 6): horses that underwent trocarization with a cecal needle (G1) or a cattle trocar (G2), and a control group (CG) without cecal trocarization. The same horses were used in each group, respecting a three-month washout period between studies. A physical examination, serial blood, and peritoneal fluid sampling were performed, prior to cecal trocarization and 2, 6 and 12 hours after the first collection and 1, 2, 3, 7, and 14 days after the procedure. Acute-phase proteins in blood and peritoneal fluid were analyzed by polyacrylamide gel electrophoresis. Horses with a high cell count in the peritoneal fluid (i.e., 10,000 cells/μl) were considered to have peritonitis and CRP if they met at least two of the following clinical criteria: anorexia, lethargy, tachycardia, tachypnea, fever, ileus, abnormal oral mucous membrane color, abnormal white blood cells count, or high blood fibrinogen concentration (> 5 g/L). All horses recovered from cecal trocarization and abdominocentesis with no major complications. Cecal trocarization caused cytologic evidence of peritonitis in G1 and G2 during the 14 days of evaluation. CRP was not observed, although a decrease in cecal motility was observed in G1 and G2 during the experimental period and three horses, one from G1 and two from G2, showed a single moment of fever. None of the groups showed leukopenia or leukocytosis, although blood neutrophil count decreased at D7 and D14 in G1 and at D14 in G2 (p ≤ 0.05). After cecal trocarization, an increase in the total nucleated cells count, total proteins, globulins, alkaline phosphatase and acute phase proteins were observed in the peritoneal fluid of G1 and G2 during the 14 days of evaluation (p ≤ 0.05), without causing clinically relevant peritonitis. Transcutaneous cecal trocarization promotes peritonitis, which is more intense with a cattle trocar than with a cecal needle. The cecal needle should be considered for cecal trocarization of horses with cecal tympany.

4 (RGSD); 2012/11621-3 (GMR); 2021/13378-8 (RGSD) for financial support and Coordenação de Aperfeiçoamento de Pessoal de Nível Superior - Brasil (CAPES) - Finance Code 001 (RGSD).

**Competing interests:** The authors have declared that no competing interests exist.

# Introduction

Horses affected by large intestine distention often present visceral pain and dyspnea associated with compartment syndrome [1, 2]. Percutaneous decompression of the cecum through the right flank is a procedure routinely performed in these cases [1–4]. Although the procedure is performed in equine practice and described in several textbooks [5–7], studies describing outcome and complications after large intestinal trocarization in equids remains underreported. After "uncomplicated" trocarization, some degree of peritonitis must be expected, and the number of nucleated cells in the peritoneal fluid may be as high as $100 \times 10^9$ /L [8, 9]. Clinical evidence of adhesion formation, abdominal abscess formation, severe intra-abdominal hemorrhage, or disseminated peritonitis have been reported in rare cases. Complications such as local cellulitis or abscess formation have been reported rarely [4, 10, 11].

A recent large retrospective study (228 cases) recommended the procedure of large intestinal trocarization in equids with colic and large intestinal gas distension. Nevertheless, more than one trocarization procedure can increase nonsurvival rates, so it should be considered only in equids for which consent for surgery was obtained [11]. Cecal trocarization has not been described as a cause of clinically relevant peritonitis (CRP) in published retrospective studies [11–15]. However, to the authors' knowledge, there are no data from controlled studies on the effect on blood and peritoneal fluid characteristics for horses that have undergone cecal trocarization or on clinical complications associated with the procedure.

Several different methods for performing cecal decompression are described in the literature [4, 10]. Instruments used for puncture vary from 15 cm, 8–16 Gauge i.v. catheters to large bore, reusable, metal trocar/cannula units (15 x 0.3 cm to 15 x 1 cm). While a large lumen ensures maximal escape of gas, it also causes more tissue damage and also causes a bigger hole for leakage [5, 9, 10]. The large size of the trocar has been questioned on the grounds that it could increase the risk of contamination of the peritoneum by leaking cecal fluid, and the use of 10-12G needles has been discouraged to avoid tearing the cecum, which could lead to leakage of ingesta into the peritoneal cavity. Nevertheless, other authors encourage use of these needles [2, 4–6, 16].

The aim of the study was to evaluate changes in the blood and peritoneal fluid of horses undergoing percutaneous cecal trocarization with a manufactured stainless steel cecal needle or a cattle trocar with cannula and to determine the incidence of peritonitis and CRP. In this study, it was hypothesized that CRP rarely occurs after percutaneous cecal trocarization, but that relevant changes in peritoneal fluid are observed directly related to the needle diameter used.

# Materials and methods

All procedures were approved by the Ethics Committee on Animal Use of Food Engineering and Animal Science, University of São Paulo, approval number ~ 13.1.450.74.4.

## Animals

Six healthy mixed breed horses (one gelding and five mares), 6–10 years of age, were included in the study. The horses had no recent history of gastrointestinal illness and were considered clinically healthy based on physical examination and routine haematological and biochemical blood profile. Horses were stabled indoors during the experimental period and were fed a maintenance diet of corn silage and grain for 15 days before the start of the experiment, to promote gas distention of the cecum. Water and trace-mineral salt were provided ad libitum.

## Transabdominal trocarization

Hair was clipped from a 10 x 10 cm diameter area over the right paralumbar fossa, and the area was aseptically prepared with povidone-iodine scrub and alcohol. The insertion site for the cecal needle or the cattle trocar with cannula was identified on the ventral side of the tuber coxae midway between the tuber coxae and the last rib. Transabdominal ultrasonography was used to determine the location and to guide the cecal needle or the cattle trocar insertion. The cecal trocarization site was sterilely infiltrated with 3 ml of 2% lidocaine solution. A 1 cm skin incision was made with a n˚ 11 scalpel blade at the trocarization site.

## Experimental protocol

The study was designed as an unblinded trial and included three experimental groups: Group 1 (G1), consisting of six horses in which cecal trocarization was performed with an atraumatic, sterile 13-cm-long, stainless-steel manufactured cecal needle with an outer diameter of 3 mm (Fig 1); Group 2 (G2): six horses in which cecal trocarization was performed using a sterile 19-cm-long stainless steel cattle trocar with an outer diameter of 6 mm and a 20-cm-long stylet (Fig 1); and a Control group (CG): six horses without trocarization of the cecum. Serial blood and peritoneal fluid samples were collected in all groups. As horses were enrolled in the study, they were randomly assigned to one of the three treatment groups, in a random crossover design. The same horses were used in each group, respecting a three-month washout period between studies.

The cecal needle or cattle trocar was directed at the contralateral olecranon and advanced into the cecum. The puncture of the cecum was guided by ultrasound imaging. Gas was aspirated for 10 minutes without the use of an extension set or suction. After this period the cecal needle or the cattle trocar was removed. The skin was not closed and povidone-iodine was applied over the wound.

Systemic antimicrobials were not administered. Dipyrone was used to control fever in horses (30 mg/kg, IV).

## Hematological parameters and abdominocentesis

Indwelling 14-gauge, 14 cm catheters were placed in the left jugular vein under aseptic conditions. Immediately before each abdominocentesis, jugular blood samples were collected using a 20ml syringe and 21G needle. After discarding the first 4 ml of blood withdrawn, the blood was transferred to four tubes (4 ml with ethylenediaminetetraacetic acid-EDTA, 4 ml without additive, 4 ml with sodium fluoride, and 4 ml with sodium citrate). Samples were stored in a cool box on ice between collection and analysis, which was performed within 60 minutes of collection.

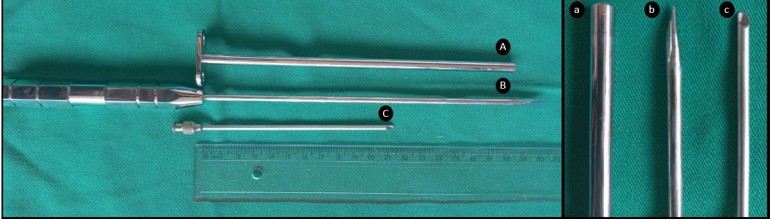

**Fig 1.** Cattle cannula (A) and stylet (B) (20 cm x 6 mm) and manufactured cecal needle (C; 13 cm x 3 mm). Note the noncutting tip cannula (a) and cutting-tip stylet (b) of cattle trocar and the Quincke point of cecal needle (c).

An area of 10 cm×10 cm on the cranial midline on the most dependent aspect of the ventral abdomen was clipped and aseptically prepared with povidone-iodine scrub and alcohol. Sonography was used to identify an area of peritoneal fluid accumulation. The subcutaneous tissue and body wall were infiltrated with 3 ml of 2% lidocaine. A stab incision was made through the skin and body wall with a number 15 blade. A gauze sponge was wrapped around a teat cannula to prevent sample contamination. The teat cannula was pushed through the body wall into the peritoneal cavity. Peritoneal fluid was collected into an EDTA tube for cytology and into sodium fluoride, sodium citrate, and no-additive tubes for biochemical parameters. Subsequent sampling was performed with a teat cannula introduced aseptically through the same skin and body wall incision for the duration of the study.

Blood collections were performed prior to cecal trocarization (T0) and 2 (T2), 6 (T6) and 12 (T12) hours after the first sampling and at day 1 (T24), day 2 (T48), day 3 (T72), day 7 (D7) and day 14 (D14) after the procedure. Peritoneal fluid sampling was always performed immediately after blood collection.

Hematological parameters included complete blood count (CBC), packed cell volume (PCV), total protein concentration (by refractometry), and plasma biochemistry. CBC variables included total white blood cell count (WBC), segmented neutrophils, lymphocytes, eosinophils, basophils, red blood cells, platelets, hemoglobin, and hematocrit. Biochemical analysis was performed on plasma and included the analytes creatinine, urea, total protein, albumin, globulin, aspartate aminotransferase, alkaline phosphatase, g-glutamyl transpeptidase, and total bilirubin.

Peritoneal fluid analysis was performed immediately after collection and included physical examination (color, appearance, and specific gravity), cellular examination (erythrocytes, and total and differential nucleated cells counts), and biochemical tests (pH, total protein, albumin, globulins, fibrinogen, alkaline phosphatase, glucose, and lactate concentrations).

All samples (blood and peritoneal fluid) were analyzed by the same automated analyzer (Mindray BC-2800 Vet®, Mindray Medical International Limited, Shenzhen, China). Blood and peritoneal fluid lactate and glucose concentrations were determined immediately after collection using the Accutrend Plus (Roche Diagnostics) and OptiumXido (Abbott Diabetes Care), respectively. A rapid heat precipitation micromethod was used for the estimation of plasma or peritoneal fluid fibrinogen.

Data were manually recorded at each measurement time point.

## Polyacrylamide gel electrophoresis

The samples were centrifuged at 1,161 g for 10 minutes and then frozen at -20˚C for further laboratory analysis. Total serum proteins were obtained by spectrophotometry in an automatic biochemical analyzer (HumaStar 300) with specific commercial reagents. For the fractionation of the different protein constituents of the serum and peritoneal fluid, electrophoresis of the respective samples was performed on polyacrylamide gel containing sodium dodecyl sulfate (SDS-PAGE), according to the technique described by Laemmli (1970) [17], with some modifications [18]. Molecular weights and concentrations of protein fractions were determined by computerized densitometry (LabImage 1D; Loccus), from scanning the gel bands. Molecular weight markers of 200, 116, 97, 66, 55, 45, 36, 29, 24 and 20 kDa (Broad Range; Bio-Rad) were used for the calculation of molecular weight, in addition of purified proteins, albumin, α1-antitrypsin, haptoglobin, ceruloplasmin, transferrin and immunoglobulin G (IgG). Using the computer software Phoresis for Windows 2000 or XP Pro (Sebia), the electrophoretic curve for each sample was displayed. Protein fractions were determined as the percentage optical absorbance, and the absolute concentration in g/dL was automatically calculated from the total

serum or peritoneal fluid protein concentration. The same operator analyzed all samples. Coefficient of variability (CV) computed as SD/mean × 100 was calculated for each protein fraction.

## Physical examination

Heart rate (HR), respiratory rate (RR), temperature (°C), and perfusion indices (mucous membrane color and capillary refill) were evaluated. Auscultation of digestive sounds (bowel movements) and behavior (appearance, appetite, sweating, kicking at abdomen, and pawing on the floor) were performed with a general score system, ranging from 0 to 3 [19]. These parameters were monitored prior to starting the cecal trocarization (T0) and 2 (T2), 6 (T6) and 12 (T12) hours after the first sampling and every 12 hours during 14 days after the procedure.

## CRP and peritonitis recording

Detection of CRP by the trocarization procedure was based on the hematological results and appearance of clinical signs, all of which were recorded. Horses were classified as having clinical signs of systemic inflammation by CRP if they met at least two of the following clinical criteria: behavior, tachycardia (i.e., heart rate > 45 beats per minute), tachypnea (i.e., respiratory rate > 25 breaths per minute), fever (i.e., rectal temperature > 38.5°C), ileus, and hyperemic or toxic (i.e., purple line at the gingival-teeth interface) mucosal color, abnormal white blood cell count ($< 5.5 \times 10^3$ WBCs/μl or $> 12.5 \times 10^3$ WBC/μl), and high blood fibrinogen concentration (> 5 g/L).

Additionally, horses had to have a high cell count in the peritoneal fluid (i.e., 10,000 cells/μl) to be classified as having peritonitis. The observed changes in acute phase proteins in plasma and peritoneal fluid were evaluated in conjunction with clinical signs and laboratory results.

## Data analysis

The experimental data underwent analysis of the normality of residues by using the Shapiro-Wilk test, followed by analysis of variance and Tukey's mean test. The non-parametric Kruskal-Wallis test was used for data that did not exhibit a normal distribution, followed by Dunn's mean rank test. The obtained data were analyzed using the SAS® University Edition 9.4 statistical software (SAS Institute, 2015). A significance level of 5% was used in all analyses.

# Results

All horses recovered without complications from cecal trocarization and abdominocentesis. Cecal trocarization caused cytologic signs of peritonitis in both experimental groups (G1 and G2), but clinically relevant peritonitis was not observed. After the cecal trocarization, there were no changes in clinical signs, except for the decrease in cecal motility observed in G1 and G2 compared with CG during the experimental period. Three horses, one from G1 (16.7%) and two from G2 (33.3%), showed a single moment of fever (T > 38.5°C) and were treated with a single dose of dipyrone.

In the hematological analysis, a decrease in some of the parameters assessed was observed over time: in G2, red cells decreased from $7.85 \pm 0.84 \times 10^6$/μL (T0) to $6.69 \pm 0.88 \times 10^6$/μL (D7) and $6.43 \pm 1.13 \times 10^6$/μL (D14); none of the groups showed leukopenia or leukocytosis over time, although in G1 neutrophil count decreased from $5.89 \pm 2 \times 10^3$ neutrophils/μL (T0) to $3.85 \pm 1.14 \times 10^3$ neutrophils/μL (D7) and $3.47 \pm 1.02 \times 10^3$ neutrophils/μL (D14); in G2,

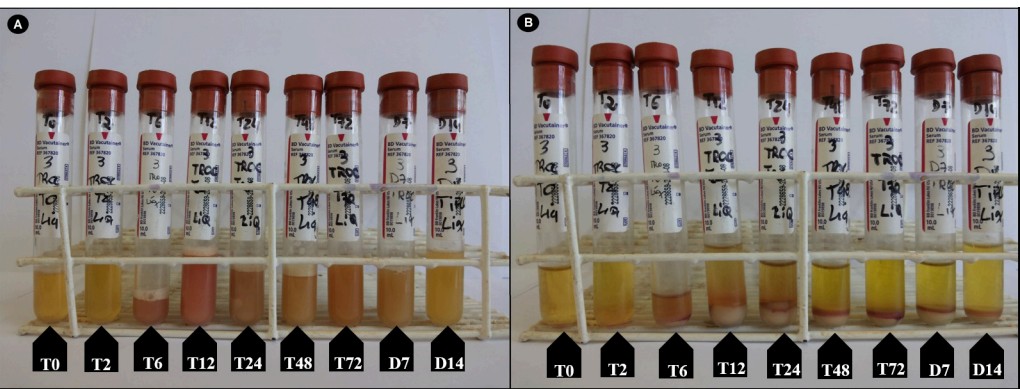

**Fig 2.** Changes over time in turbidity and color of the peritoneal fluid of a horse after cecal trocarization with a cattle trocar (A). Sediment is observed after centrifugation (B).

the neutrophil count decreased from $4.66 \pm 1.74 \times 10^3$ neutrophils/ μL (T0) to $2.88 \pm 1.27 \times 10^3$ neutrophils/ μL (D14). No other changes were observed in the hematological analysis. Serum biochemical analysis revealed no changes over time or between experimental groups.

Important changes over time were observed in peritoneal fluid collections after trocarization. In G1 and G2, increased turbidity and a change in color from light yellow to orange were observed from T2 to D14 (Fig 2A). Sediment was observed in some tubes from G2 after centrifugation (Fig 2B).

On cytological evaluation of the peritoneal fluid, a progressive increase in the total number of nucleated cells was observed in G1 and G2 from T6 to D14. The total number of nucleated cells was higher in G1 compared with CG from T6 to D14 and in G2 compared with CG from T2 to D14. Higher peritoneal cellularity was observed at T12 and T48 in G2 compared with G1 (Table 1 and Fig 3).

Increased segmented neutrophils were observed from T6 to D14 in G1 and from T2 to D14 in G2. The number of neutrophils was higher in G1 and G2 than in CG from T6 to D14 (Table 2). Increased macrophages were observed from T48 to D14 in CG, in T24, T72, D7, and

**Table 1. Mean and standard deviation (SD) of total nucleated cells count in the peritoneal fluid of healthy horses (CG) and horses submitted to cecal trocarization with a cecal needle (G1) or a cattle trocar (G2).**

| Total Nucleated Cells ($\times 10^3$/μL) | | | |
|---|---|---|---|
| Time | CG | G1 | G2 |
| T0 | $2,72 \pm 1,42^A$ | $3,12 \pm 1,83^A$ | $2,62 \pm 1,24^A$ |
| T2 | $2,41 \pm 1,18^B$ | $4,00 \pm 2,51^{AB}$ | $6,73 \pm 4,99^A$ |
| T6 | $2,58 \pm 1,77^B$ | $9,32 \pm 7,12^{*A}$ | $21,50 \pm 20,24^{*A}$ |
| T12 | $2,78 \pm 1,72^C$ | $8,13 \pm 3,56^{*B}$ | $63,88 \pm 68,50^{*A}$ |
| T24 | $3,69 \pm 2,66^B$ | $13,85 \pm 8,57^{*A}$ | $73,38 \pm 78,54^{*A}$ |
| T48 | $2,99 \pm 1,68^C$ | $11,26 \pm 7,34^{*B}$ | $50,96 \pm 43,73^{*A}$ |
| T72 | $2,48 \pm 1,14^B$ | $19,08 \pm 4,51^{*A}$ | $66,97 \pm 67,73^{*A}$ |
| D7 | $2,68 \pm 1,06^B$ | $41,08 \pm 35,10^{*A}$ | $69,38 \pm 32,91^{*A}$ |
| D14 | $2,08 \pm 1,27^B$ | $30,31 \pm 23,46^{*A}$ | $61,67 \pm 47,26^{*A}$ |
| Mean±SD | $2,71 \pm 1,55^C$ | $15,57 \pm 17,46^B$ | $46,34 \pm 51,39^A$ |

(*) represent differences compared to baseline time (T0) within the group; Different capital letters represent differences between groups at the same time; $p \leq 0.05$.
Reference value: 3,000–5,000 cells/μL [20]

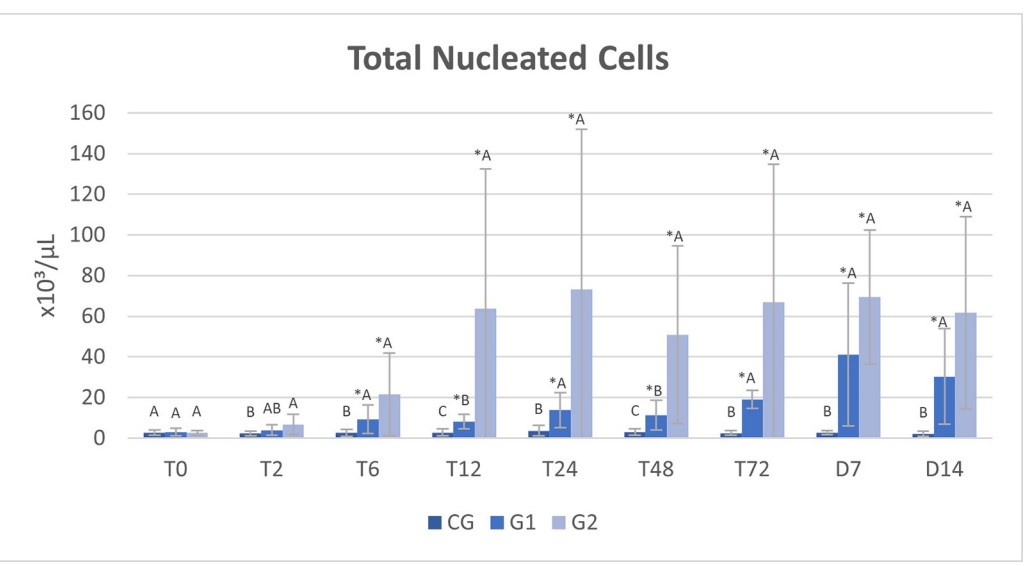

**Fig 3.** Mean and standard deviation (SD) of total nucleated cells count in the peritoneal fluid of healthy horses (CG) and horses submitted to cecal trocarization with a cecal needle (G1) or a cattle trocar (G2).

D14 in G1, and from T12 to D14 in G2 (Table 3). Increased lymphocytes were observed at T12, T72, and D7 in G1 and from T12 to D14 in G2. G2 presented higher lymphocytes from T12 to D14 than CG, and G1 presented higher lymphocytes from T72 to D14 than CG (Table 4).

Biochemical analysis of peritoneal fluid showed an increase in total proteins and globulins over time, from T12 to D14, in G1 and G2 (Tables 5, 6 and Fig 4). Alkaline phosphatase increased at D7 in G1 and from T6 to D14 in G2 (Table 7 and Fig 5). No other biochemical changes were observed in the peritoneal fluid.

Albumin, globulins alpha-1 acid glycoprotein, alpha-1 antitrypsin, haptoglobin, transferrin, ceruloplasmin, and immunoglobulins IgA and IgG were detected by electrophoretic fractionation in polyacrylamide gel in plasma and peritoneal fluid of the studied horses.

**Table 2. Mean and standard deviation (SD) of segmented neutrophils in the peritoneal fluid of healthy horses (CG) and horses submitted to cecal trocarization with a cecal needle (G1) or a cattle trocar (G2).**

| | Segmented Neutrophils ($\times10^3/\mu L$) | | |
|---|---|---|---|
| **Time** | **CG** | **G1** | **G2** |
| T0 | $2,07 \pm 1,25^A$ | $2,11 \pm 1,68^A$ | $0,83 \pm 0,55^A$ |
| T2 | $1,88 \pm 0,00^A$ | $2,96 \pm 1,85^A$ | $5,34 \pm 5,40^{*A}$ |
| T6 | $1,94 \pm 1,75^B$ | $6,65 \pm 4,46^{*A}$ | $18,99 \pm 19,30^{*A}$ |
| T12 | $1,9 \pm 1,36^B$ | $6,28 \pm 3,26^{*A}$ | $55,73 \pm 67,28^{*A}$ |
| T24 | $2,71 \pm 2,72^B$ | $10,1 \pm 7,74^{*A}$ | $52,21 \pm 53,80^{*A}$ |
| T48 | $1,55 \pm 1,27^B$ | $8,77 \pm 6,74^{*A}$ | $36,86 \pm 33,67^{*A}$ |
| T72 | $1,08 \pm 0,71^B$ | $13,75 \pm 3,47^{*A}$ | $50,22 \pm 51,97^{*A}$ |
| D7 | $1,42 \pm 0,84^B$ | $29,66 \pm 24,22^{*A}$ | $57,33 \pm 30,37^{*A}$ |
| D14 | $0,62 \pm 0,55^{*B}$ | $23,1 \pm 18,92^{*A}$ | $52,04 \pm 43,28^{*A}$ |
| Mean ±SD | $1,69 \pm 1,42^C$ | $11,49 \pm 12,82^B$ | $36,62 \pm 42,02^A$ |

(*) represent differences compared to baseline time (T0) within the group; Different capital letters represent differences between groups at the same time; $p \leq 0.05$.

**Table 3. Mean and standard deviation (SD) of macrophages in the peritoneal fluid of healthy horses (CG) and horses submitted to cecal trocarization with a cecal needle (G1) or a cattle trocar (G2).**

| Macrophages (x10³/μL) | | | |
|---|---|---|---|
| Time | CG | G1 | G2 |
| T0 | 0,32 ± 0,29 | 0,72 ± 0,75 | 1,17 ± 0,91 |
| T2 | 0,12 ± 0,21 | 0,61 ± 1,02 | 1,10 ± 0,62 |
| T6 | 0,16 ± 0,20 | 2,29 ± 3,99 | 1,78 ± 1,00 |
| T12 | 0,49 ± 0,60 | 0,90 ± 1,37 | 5,00 ± 3,69* |
| T24 | 1,02 ± 1,08 | 2,84 ± 2,33* | 18,28 ± 25,50* |
| T48 | 1,26 ± 0,97* | 1,83 ± 1,45 | 11,04 ± 8,58* |
| T72 | 1,24 ± 0,79* | 3,44 ± 1,48* | 11,96 ± 12,39* |
| D7 | 0,96 ± 0,35* | 9,89 ± 11,91* | 8,45 ± 2,24* |
| D14 | 1,24 ± 0,79* | 5,87 ± 4,82* | 6,80 ± 4,19* |
| Mean ±SD | 0,75 ± 0,76[C] | 3,15 ± 4,91[B] | 7,29 ± 10,67[A] |

(*) represent differences compared to baseline time (T0) within the group; Different capital letters represent differences between groups at the same time; p≤0.05.

In the analysis of changes in the expression of serum acute phase proteins, only ceruloplasmin showed an increase over time. Ceruloplasmin increased at D7 in CG, at T12, T48, D7, and D14 in G1, and at T6, T12, D7, and D14 in G2 (Table 8).

In the peritoneal fluid, no changes in acute-phase proteins were observed over time in the control group. An increase in albumin was observed from T24 to D14 in G1 and from T6 to D14 in G2. Higher mean values for albumin were observed in G2 than in G1 and CG, and G1 had higher mean values for albumin than CG (Table 9). Increased IgA levels were observed from T12 to D14 in G1 and from T6 to D14 in G2. CG showed lower IgA concentrations than G1 from T12 to D14 and lower IgA concentrations than G2 from T6 to D14 (Table 10). Increased IgG levels were observed from T12 to D14 in G1 and at T6, T48, T72, and D7 in G2. CG showed lower mean IgG levels than GA and GT (Table 11). Increased transferrin levels were observed from T12 to D14 in G1 and at T6, T12, T24, T48, D7, and D14 in G2. GC had a presented lower mean transferrin concentration than G1 and G2, and G1 had a lower mean transferrin concentration than G2 (Table 12). Increased haptoglobin levels were observed

**Table 4. Mean and standard deviation (SD) of lymphocytes in the peritoneal fluid of healthy horses (CG) and horses submitted to cecal trocarization with a cecal needle (G1) or a cattle trocar (G2).**

| Lymphocytes (x10³/μL) | | | |
|---|---|---|---|
| Time | CG | G1 | G2 |
| T0 | 0,32 ± 0,30[A] | 0,29 ± 0,25 [A] | 0,61 ± 1,09[A] |
| T2 | 0,42 ± 0,34[A] | 0,42 ± 0,24[A] | 0,25 ± 0,12[A] |
| T6 | 0,44 ± 0,39[A] | 0,34 ± 0,22[A] | 0,58 ± 0,37[A] |
| T12 | 0,47 ± 0,47[B] | 0,81 ± 0,52*[AB] | 1,74 ± 1,17*[A] |
| T24 | 0,23 ± 0,19[B] | 0,77 ± 0,48[AB] | 1,82 ± 1,50*[A] |
| T48 | 0,22 ± 0,17[B] | 0,36 ± 0,24[AB] | 1,01 ± 0,78*[A] |
| T72 | 0,13 ± 0,11[B] | 0,87 ± 0,52*[A] | 2,18 ± 2,91*[A] |
| D7 | 0,26 ± 0,24[B] | 0,86 ± 0,78*[A] | 2,52 ± 1,62*[A] |
| D14 | 0,13 ± 0,09[B] | 1,22 ± 1,46[A] | 2,27 ± 2,15*[A] |
| Mean ±SD | 0,29 ± 0,29[C] | 0,66 ± 0,66[B] | 1,44 ± 1,61[A] |

(*) represent differences compared to baseline time (T0) within the group; Different capital letters represent differences between groups at the same time; p≤0.05.

**Table 5. Mean and standard deviation (SD) of total proteins in the peritoneal fluid of healthy horses (CG) and horses submitted to cecal trocarization with a cecal needle (G1) or a cattle trocar (G2).**

| | Total proteins (g/dL) | | |
|---|---|---|---|
| Time | CG | G1 | G2 |
| T0 | 1,50 ± 0,60 | 1,46 ± 0,45 | 1,69 ± 0,48 |
| T2 | 1,74 ± 0,84 | 1,57 ± 0,45 | 2,04 ± 0,32 |
| T6 | 1,36 ± 0,48 | 1,75 ± 0,36 | 2,75 ± 0,54 |
| T12 | 1,47 ± 0,47 | 2,24 ± 0,42* | 3,55 ± 0,67* |
| T24 | 1,44 ±0,42 | 2,62 ± 0,55* | 3,67 ± 1,06* |
| T48 | 1,31 ± 0,38 | 2,57 ± 0,92* | 3,13 ± 0,76* |
| T72 | 1,30 ±0,35 | 2,66 ± 0,83* | 3,31 ± 0,86* |
| D7 | 1,61 ± 0,48 | 2,97 ± 1,26* | 3,54 ± 0,74* |
| D14 | 1,53 ± 0,55 | 3,08 ± 1,38* | 3,26 ± 1,00* |
| Mean ±SD | 1,47 ± 0,50[C] | 2,32 ± 0,92[B] | 2,99 ± 0,95[A] |

(*) represent differences compared to baseline time (T0) within the group; Different capital letters represent differences between groups at the same time; p≤0.05.
Reference values: 1–2,4 g/dL [20]

from T24 to D14 in G1 and from T6 to D14 in G2. CG had lower haptoglobin concentrations than G1 from T24 to D14 and lower than G2 from T6 to D14 (Table 13).

## Discussion

Previously, only a few original papers have been published on transcutaneous cecal decompression and only one complication has been described [4, 11, 21, 22]. The instruments used for puncture vary from different types of cattle trocars with cannula to different sizes of over-the-needle i.v. catheters. In the present study, complications following transabdominal cecal trocarization with a cattle trocar and cannula (outer diameter of 6 mm) were compared with a manufactured, reusable cecal needle, with a similar size of 12 G i.v. catheter (outer diameter of 3 mm).

Transcutaneous decompression of the cecum can lead to potentially life-threatening complications such as abscess formation, septic peritonitis, and hemorrhage [10]. A recent study

**Table 6. Mean and standard deviation (SD) of globulins in the peritoneal fluid of healthy horses (CG) and horses submitted to cecal trocarization with a cecal needle (G1) or a cattle trocar (G2).**

| | Globulins (mg/dL) | | |
|---|---|---|---|
| Time | CG | G1 | G2 |
| T0 | 0,66 ± 0,30 | 0,70 ± 0,19 | 0,75 ± 0,27 |
| T2 | 0,96 ± 0,78 | 0,73 ± 0,20 | 0,96 ± 0,23 |
| T6 | 0,59 ± 0,23 | 0,87 ± 0,19 | 1,37 ± 0,37 |
| T12 | 0,64 ± 0,22 | 1,17 ± 0,24* | 1,77 ± 0,47* |
| T24 | 0,62 ± 0,20 | 1,35 ± 0,28* | 1,88 ± 0,63* |
| T48 | 0,57 ± 0,20 | 1,37 ± 0,55* | 1,60 ± 0,49* |
| T72 | 0,51 ± 0,26 | 1,39 ± 0,53* | 1,68 ± 0,58* |
| D7 | 0,69 ± 0,26 | 1,50 ± 0,70* | 1,83 ± 0,51* |
| D14 | 0,67 ± 0,33 | 1,60 ± 0,81* | 1,71 ± 0,67* |
| Mean ±SD | 0,66 ± 0,35[C] | 1,19 ± 0,52[B] | 1,50 ± 0,58[A] |

(*) represent differences compared to baseline time (T0) within the group; Different capital letters represent differences between groups at the same time; p≤0.05.
Reference values: 0,7–1,4 g/dL [20]

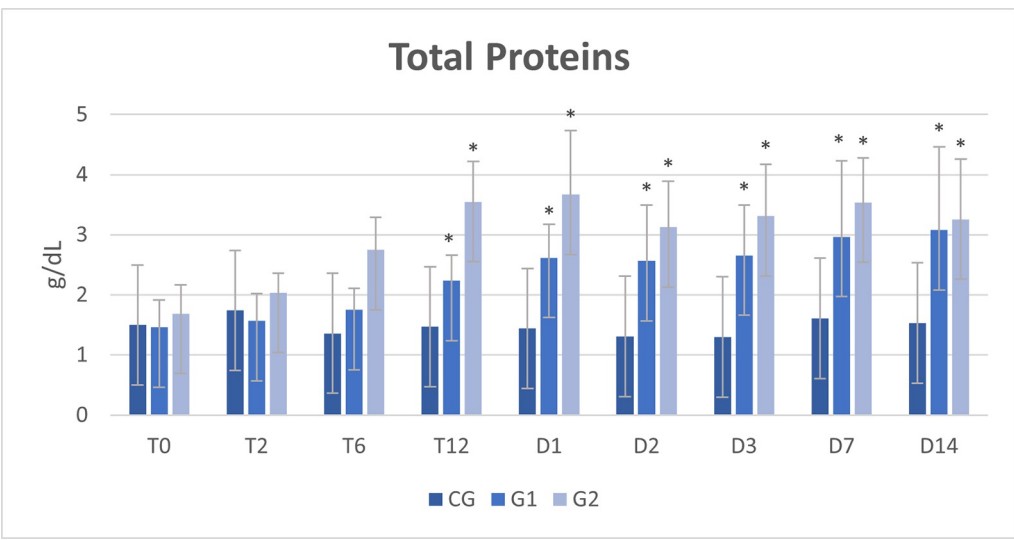

**Fig 4.** Mean and standard deviation (SD) of total proteins in the peritoneal fluid of healthy horses (CG) and horses submitted to cecal trocarization with a cecal needle (G1) or a cattle trocar (G2).

has shown that large intestinal trocarization resulted in few clinically apparent short-term adverse reactions. Abnormal peritoneal fluid values were common in horses after large intestinal trocarization. However, clinically relevant peritonitis was rare and not related to the number or location of trocarization procedures [11]. In the present study, cecal trocarization with a cecal needle or a cattle trocar caused significant changes in peritoneal fluid, but clinically relevant peritonitis was not observed. The self-limiting, localized effect of peritonitis has been suggested as an explanation for a lack of systemic problems in most cases [9].

Transcutaneous cecal trocarization resulted in a decrease in cecal motility over time, possibly due to local inflammation caused by the puncture. Fever was observed in a higher percentage of horses in G2 (33.3%) compared with G1 (16.7%), indicating a greater systemic impact of cecal trocatization with a cattle trocar. Hematological evaluation showed migration of

**Table 7. Mean and standard deviation (SD) of alkaline phosphatase in the peritoneal fluid of healthy horses (CG) and horses submitted to cecal trocarization with a cecal needle (G1) or a cattle trocar (G2).**

| Alkaline Phosphatase (mg/dL) | | | |
|---|---|---|---|
| **Time** | **CG** | **G1** | **G2** |
| **T0** | 29,67 ± 21,59 | 33,17 ± 22,84 | 25,20 ± 11,76 |
| **T2** | 29,17 ± 19,03 | 85,00 ± 68,60 | 200,40 ± 94,93 |
| **T6** | 28,50 ± 17,74 | 306,00 ± 244,96 | 1202,00 ± 432,90* |
| **T12** | 29,00 ± 18,13 | 570,17 ± 567,38 | 2521,40 ± 1509,73* |
| **T24** | 28,67 ± 18,01 | 722,83 ± 616,70 | 1762,60 ± 967,72* |
| **T48** | 23,50 ± 12,39 | 658,60 ± 517,64 | 1893,80 ± 1113,17* |
| **T72** | 21,67 ± 11,43 | 849,40 ± 633,33 | 1951,20 ± 1237,30* |
| **D7** | 35,00 ± 23,47 | 1520,60 ± 1773,79* | 2047,00 ± 1362,59* |
| **D14** | 26,33 ± 15,31 | 874,00 ± 631,07 | 1430,00 ± 1123,88* |
| **Mean ±SD** | 27,94 ± 16,84[C] | 624,42 ± 779,40[B] | 1448,18 ± 1222,21[A] |

(*) represent differences compared to baseline time (T0) within the group; Different capital letters represent differences between groups at the same time; $p \leq 0.05$.
Reference values: 0–161 U/L [20]

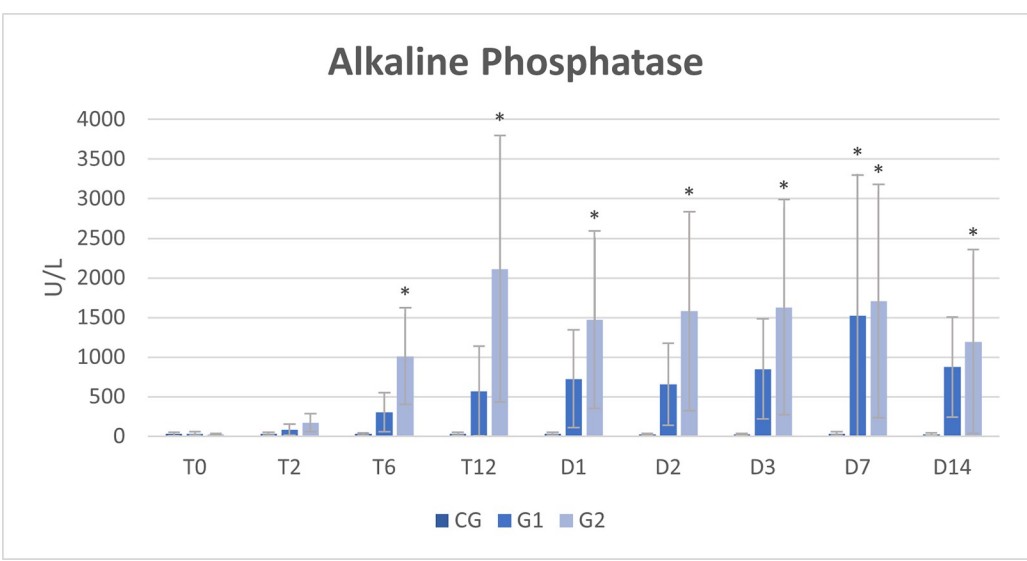

**Fig 5.** Mean and standard deviation (SD) of alkaline phosphatase in the peritoneal fluid of healthy horses (CG) and horses submitted to cecal trocarization with a cecal needle (G1) or a cattle trocar (G2).

leukocytes from the bloodstream to the abdominal cavity after cecal trocarization [23, 24], as evidenced by a decrease of neutrophils count in the blood and a progressive increase in the total number of nucleated cells count in peritoneal fluid, including neutrophils, macrophages, and lymphocytes. Serial abdominocentesis was not responsible for hematological or peritoneal fluid alterations, as evidenced in CG.

It is known that cecal puncture may promote some degree of leakage of cecal content into the peritoneal cavity and represents a peritoneal insult [25]. Cecal trocarization promoted increase in total proteins, globulins, total nucleated cells count and alkaline phosphatase in the peritoneal fluid that was far above the physiological level for this species [26]. The increase in proteins in peritoneal fluid occurs when there is an inflammatory process that increases the peritoneal permeability [24, 27]. Migration of neutrophils into the peritoneal cavity as found in this study occurs due to the local release of chemotactic substances [28], which act in the

**Table 8. Mean and standard deviation (SD) of serum ceruloplasmin of healthy horses (CG) and horses submitted to cecal trocarization with a cecal needle (G1) or a cattle trocar (G2).**

| Ceruloplasmin (mg/dL) | | | |
|---|---|---|---|
| Time | CG | G1 | G2 |
| T0 | 5,90 ± 5,18 | 9,17 ± 2,82 | 13,02 ± 8,43 |
| T2 | 5,48 ± 1,79 | 9,42 ± 6,07 | 16,41 ± 9,09 |
| T6 | 2,38 ± 1,13 | 9,61 ± 3,63 | 22,03 ± 6,09* |
| T12 | 6,10 ± 5,67 | 13,72 ± 4,50* | 16,93 ± 7,96* |
| T24 | 9,57 ± 3,70 | 12,11 ± 4,63 | 17,09 ± 4,90 |
| T48 | 9,69 ± 4,43 | 13,78 ± 4,08* | 18,81 ± 10,65 |
| T72 | 8,02 ± 6,96 | 12,05 ± 3,01 | 22,16 ± 8,01 |
| D7 | 11,41 ± 5,24* | 17,76 ± 8,85* | 25,90 ± 5,43* |
| D14 | 8,94 ± 1,72 | 16,13 ± 7,22* | 24,34 ± 8,41* |
| Mean ±SD | 7,50 ± 4,85[C] | 12,64 ± 5,71[B] | 19,63 ± 8,17[A] |

(*) represent differences compared to baseline time (T0) within the group; Different capital letters represent differences between groups at the same time; p≤0.05.

**Table 9. Mean and standard deviation (SD) of albumin in the peritoneal fluid of healthy horses (CG) and horses submitted to cecal trocarization with a cecal needle (G1) or a cattle trocar (G2).**

| Albumin (mg/dL) | | | |
|---|---|---|---|
| Time | CG | G1 | G2 |
| T0 | 1017,57 ± 387,28 | 1039,31 ± 347,99 | 1177,93 ± 367,50 |
| T2 | 1184,33 ± 562,64 | 1074,29 ± 344,99 | 1384,48 ± 197,07 |
| T6 | 920,54 ± 326,14 | 1161,28 ± 251,51 | 1824,18 ± 394,56* |
| T12 | 1006,54 ± 305,58 | 1488,13 ± 346,72 | 2335,49 ± 507,02* |
| T24 | 978,01 ± 292,14 | 1601,55 ± 481,16* | 2432,33 ± 846,07* |
| T48 | 897,28 ± 243,29 | 1610,96 ± 509,95* | 2027,85 ± 534,99* |
| T72 | 874,96 ± 233,97 | 1608,07 ± 388,60* | 2156,67 ± 630,56* |
| D7 | 1112,85 ± 320,44 | 1883,50 ± 674,01* | 2310,53 ± 491,75* |
| D14 | 1042,70 ± 352,00 | 1991,77 ± 860,75* | 2084,38 ± 764,62* |
| Mean ±SD | 1003,87 ± 335,09[C] | 1495,43 ± 550,34[B] | 1970,43 ± 653,33[A] |

(*) represent differences compared to baseline time (T0) within the group; Different capital letters represent differences between groups at the same time; p≤0.05.

primary cellular defense against microorganisms and have intense phagocytic activity [24]. Moreover, the increase of alkaline phosphatase enzyme in peritoneal fluid observed in this study was likely related to induction of an inflammatory processes due to trocarization and potentially the presence of intestinal bacteria in the fluid [29, 30]. All these changes characterize the occurrence of peritonitis in horses submitted to cecal trocarization, which was present throughout the 14 days of the study evaluation. Therefore, the results of this study show that serial abdominocentesis is not responsible for the increase in total proteins, total nucleated cells, and alkaline phosphatase in the peritoneal fluid.

Cecal trocarization with a cattle trocar, showed higher total proteins, globulins, total nucleated cells count and alkaline phosphatase than cecal trocarization with a cecal needle. This fact indicates that the diameter of the material used to perform cecal trocarization is related to the magnitude of peritonitis. It is noteworthy that all horses in the study had no other changes on physical examination and did not require antimicrobial therapy or supportive care, and there were no major complications or deaths, although they did have significant changes in

**Table 10. Mean and standard deviation (SD) of IgA in the peritoneal fluid of healthy horses (CG) and horses submitted to cecal trocarization with a cecal needle (G1) or a cattle trocar (G2).**

| IgA (mg/dL) | | | |
|---|---|---|---|
| Time | CG | G1 | G2 |
| T0 | 9,93 ± 4,04[A] | 9,90 ± 6,04[A] | 12,91 ± 9,08[A] |
| T2 | 10,70 ± 4,95[A] | 11,73 ± 7,01[A] | 24,79 ± 14,71[A] |
| T6 | 8,88 ± 3,84[B] | 18,26 ± 9,16[AB] | 30,55 ± 12,87[*A] |
| T12 | 9,67 ± 4,43[B] | 28,83 ± 11,79[*A] | 30,38 ± 15,09[*A] |
| T24 | 8,74 ± 3,48[B] | 36,76 ± 10,69[*A] | 32,16 ± 13,31[*A] |
| T48 | 7,64 ± 4,99[B] | 45,82 ± 23,52[*A] | 36,08 ± 20,47[*A] |
| T72 | 8,67 ± 3,80[B] | 50,35 ± 32,30[*A] | 33,71 ± 8,66[*A] |
| D7 | 10,99 ± 7,19[B] | 36,02 ± 19,96[*A] | 31,22 ± 10,40[*A] |
| D14 | 10,50 ± 6,67[B] | 31,17 ± 18,16[*A] | 27,59 ± 10,62[*A] |
| Mean ±SD | 9,53 ± 4,71[B] | 29,87 ± 20,41[A] | 28,82 ± 13,64[A] |

(*) represent differences compared to baseline time (T0) within the group; Different capital letters represent differences between groups at the same time; p≤0.05.

**Table 11. Mean and standard deviation (SD) of IgG in the peritoneal fluid of healthy horses (CG) and horses submitted to cecal trocarization with a cecal needle (G1) or a cattle trocar (G2).**

| | IgG (mg/dL) | | |
|---|---|---|---|
| Time | CG | G1 | G2 |
| T0 | 125,31 ± 75,20 | 107,28 ± 18,63 | 160,17 ± 73,27 |
| T2 | 170,90 ± 138,50 | 149,33 ± 77,12 | 202,71 ± 76,83 |
| T6 | 117,46 ± 66,60 | 179,51 ± 79,42 | 249,31 ± 43,99* |
| T12 | 142,67 ± 86,14 | 227,30 ± 84,71* | 247,14 ± 46,13 |
| T24 | 128,24 ± 60,89 | 303,32 ± 82,76* | 227,43 ± 86,43 |
| T48 | 126,81 ± 70,22 | 356,22 ± 170,70* | 276,55 ± 87,28* |
| T72 | 123,96 ± 67,83 | 403,06 ± 221,10* | 305,42 ± 56,36* |
| D7 | 133,17 ± 62,38 | 406,32 ± 186,37* | 315,62 ± 44,32* |
| D14 | 133,41 ± 65,29 | 351,24 ± 180,92* | 243,28 ± 58,04 |
| Mean ±SD | 133,55 ± 56,91[B] | 275,95 ± 114,00[A] | 247,52 ± 75,41[A] |

(*) represent differences compared to baseline time (T0) within the group; Different capital letters represent differences between groups at the same time; $p \leq 0.05$.

peritoneal fluid. The use of antibiotics after large intestinal trocarization is rarely justified, and the antimicrobial class and dose administered are often not reported [5]. Some authors recommend the use of systemic therapy with antimicrobials only in patients with evidence of complications [31]. Several authors recommend local administration of antimicrobials before or during trocar removal [7, 32], but this procedure was not performed in this study.

Acute-phase proteins obtained by polyacrylamide gel electrophoresis separate protein into six fractions, albumin, α1, α2, β1, β2, and γ-globulins in mammals, resulting in a typical electrophoretic pattern for protein distribution [33, 34]. Abnormalities in the electrophoretic pattern are mainly associated with infectious/inflammatory diseases [35]. Cecal trocarization promoted the increase in serum ceruloplasmin, an acute-phase protein, during 14 days of evaluation, as observed in horses with acute abdomen undergoing exploratory laparotomy [18].

In the peritoneal fluid, the increase in total proteins and globulins observed during the 14 days of evaluation could be explained by the increase in the concentration of albumin, IgA, IgG, transferrin, and haptoglobin. Acute-phase proteins are synthesized predominantly in the

**Table 12. Mean and standard deviation (SD) of transferrin in the peritoneal fluid of healthy horses (CG) and horses submitted to cecal trocarization with a cecal needle (G1) or a cattle trocar (G2).**

| | Transferrin (mg/dL) | | |
|---|---|---|---|
| Time | CG | G1 | G2 |
| T0 | 83,54 ± 32,34 | 75,23 ± 27,41 | 103,93 ± 31,65 |
| T2 | 89,99 ± 44,26 | 79,18 ± 30,06 | 136,20 ± 43,60 |
| T6 | 77,98 ± 30,25 | 85,92 ± 28,89 | 196,16 ± 59,00* |
| T12 | 85,26 ± 26,38 | 120,92 ± 41,11* | 301,41 ± 120,55* |
| T24 | 80,83 ± 24,63 | 157,97 ± 58,21* | 274,07 ± 114,10* |
| T48 | 71,70 ± 19,05 | 166,89 ± 122,95* | 204,74 ± 82,68* |
| T72 | 68,43 ± 28,59 | 169,82 ± 115,60* | 174,42 ± 64,73 |
| D7 | 98,54 ± 34,15 | 185,05 ± 118,65* | 237,57 ± 68,79* |
| D14 | 88,80 ± 35,83 | 207,25 ± 138,74* | 268,99 ± 93,72* |
| Mean ±SD | 82,79 ± 30,24[C] | 138,69 ± 89,65[B] | 210,83 ± 96,33[A] |

(*) represent differences compared to baseline time (T0) within the group; Different capital letters represent differences between groups at the same time; $p \leq 0.05$.

**Table 13. Mean and standard deviation (SD) of haptoglobin in the peritoneal fluid of healthy horses (CG) and horses submitted to cecal trocarization with a cecal needle (G1) or a cattle trocar (G2).**

| Time | Haptoglobin (mg/dL) | | |
|------|------|------|------|
|      | CG | G1 | G2 |
| T0 | $6,62 \pm 3,90^A$ | $5,38 \pm 2,13^A$ | $8,31 \pm 7,81^A$ |
| T2 | $8,73 \pm 5,17^A$ | $4,70 \pm 2,73^A$ | $12,38 \pm 6,33^A$ |
| T6 | $6,37 \pm 3,16^B$ | $7,85 \pm 2,93^{AB}$ | $35,67 \pm 31,57^{*A}$ |
| T12 | $6,46 \pm 2,72^B$ | $11,70 \pm 9,33^{AB}$ | $55,72 \pm 31,98^{*A}$ |
| T24 | $5,88 \pm 2,58^B$ | $24,37 \pm 17,48^{*A}$ | $67,74 \pm 47,64^{*A}$ |
| T48 | $4,98 \pm 1,77^B$ | $24,57 \pm 32,84^{*A}$ | $56,86 \pm 47,60^{*A}$ |
| T72 | $4,97 \pm 1,43^B$ | $23,04 \pm 28,43^{*A}$ | $73,10 \pm 53,08^{*A}$ |
| D7 | $7,18 \pm 3,88^B$ | $47,27 \pm 80,23^{*A}$ | $86,86 \pm 67,37^{*A}$ |
| D14 | $5,53 \pm 2,41^B$ | $32,53 \pm 55,94^{*A}$ | $54,11 \pm 42,13^{*A}$ |
| Mean ±SD | $6,30 \pm 3,15^C$ | $20,16 \pm 28,63^B$ | $50,08 \pm 45,59^A$ |

($^*$) represent differences compared to baseline time (T0) within the group; Different capital letters represent differences between groups at the same time; p≤0.05.

liver in response to secretion of pro-inflammatory cytokines. Often these proteins increase because of activation of the inflammatory response to regulate the different phases of inflammation [36, 37]. It is possible that ceruloplasmin, albumin, IgA, IgG, transferrin, and haptoglobin are the acute-phase proteins most responsive to abdominal insults such as peritoneal inflammation and infection promoted by the cecal trocarization [38–41], since the horses in the control group showed no changes in acute-phase proteins in plasma and peritoneal fluid over time. Similarly, other studies have shown an increase in fibrinogen and haptoglobin in the peritoneal fluid of horses with peritonitis [12, 27, 40, 42–45]. Cecal trocarization with a cattle trocar resulted in higher levels of acute phase proteins in the peritoneal fluid than a cecal needle, demonstrating that an inflammatory insult is present and may be influenced by the size of the needle. In veterinary medicine, serum protein electrophoresis is recognized as a useful tool for the diagnosis, prognosis, and monitoring of several diseases [46].

The main limitation of the present study was the small number of healthy horses used, a fact that increases individual variation. It is also important to note that these were healthy horses, without a severely distended cecum. In horses with colic, where trocarization of the cecum is considered necessary, would potentially have more leakage from around the trocar/needle following puncture since the cecum is under more pressure. Another limitation was that no long-term follow-up was performed to determine whether complications such as intra-abdominal abscess formation or adhesions occurred; therefore, long-term complications cannot be excluded.

## Conclusion

Transcutaneous cecal trocarization promotes peritonitis, as evidenced by migration of neutrophils from the bloodstream into the peritoneal fluid, peritoneal increase in total nucleated cells count, total proteins, alkaline phosphatase, and acute phase proteins, without causing clinically relevant peritonitis. The cattle trocar caused more severe peritonitis than the cecal needle. Cecal trocarization with a cecal needle should be considered for horses with cecal gas distention.

## Author Contributions

**Conceptualization:** Renata Gebara Sampaio Dória.

**Data curation:** Renata Gebara Sampaio Dória, Gustavo Morandini Reginato, Yumi de Barcelos Hayasaka, Paulo Fantinato Neto, Danielle Passarelli, Julia de Assis Arantes.

**Formal analysis:** Renata Gebara Sampaio Dória, Gustavo Morandini Reginato, Paulo Fantinato Neto, Danielle Passarelli, Julia de Assis Arantes.

**Funding acquisition:** Renata Gebara Sampaio Dória, Gustavo Morandini Reginato.

**Investigation:** Renata Gebara Sampaio Dória, Gustavo Morandini Reginato, Yumi de Barcelos Hayasaka.

**Methodology:** Renata Gebara Sampaio Dória, Gustavo Morandini Reginato, Yumi de Barcelos Hayasaka, Paulo Fantinato Neto, Danielle Passarelli, Julia de Assis Arantes.

**Project administration:** Renata Gebara Sampaio Dória.

**Resources:** Renata Gebara Sampaio Dória.

**Supervision:** Renata Gebara Sampaio Dória.

**Validation:** Renata Gebara Sampaio Dória.

**Visualization:** Renata Gebara Sampaio Dória.

**Writing – original draft:** Renata Gebara Sampaio Dória, Gustavo Morandini Reginato.

**Writing – review & editing:** Yumi de Barcelos Hayasaka, Julia de Assis Arantes.

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
