## [Decision Letter · Decision Letter 0]

14 Sep 2022

PONE-D-22-22467Complications following transcutaneous cecal trocarization in horses with cattle trocar and cecal needlePLOS ONE

Dear Dr. Dória,

Thank you for submitting your manuscript to PLOS ONE. After careful consideration, we feel that it has merit but does not fully meet PLOS ONE’s publication criteria as it currently stands. Therefore, we invite you to submit a revised version of the manuscript that addresses the points raised during the review process.

A marked-up copy of your manuscript that highlights changes made to the original version. You should upload this as a separate file labeled 'Revised Manuscript with Track Changes'.An unmarked version of your revised paper without tracked changes. You should upload this as a separate file labeled 'Manuscript'.If applicable, we recommend that you deposit your laboratory protocols in protocols.io to enhance the reproducibility of your results. Protocols.io assigns your protocol its own identifier (DOI) so that it can be cited independently in the future. For instructions see: https://journals.plos.org/plosone/s/submission-guidelines#loc-laboratory-protocols. Additionally, PLOS ONE offers an option for publishing peer-reviewed Lab Protocol articles, which describe protocols hosted on protocols.io. Read more information on sharing protocols at https://plos.org/protocols?utm_medium=editorial-email&utm_source=authorletters&utm_campaign=protocols.

We look forward to receiving your revised manuscript.

Kind regards,

Benito Soto-Blanco, DVM, MSc, PhD

Academic Editor

PLOS ONE

2. As part of your revision, please complete and submit a copy of the Full ARRIVE 2.0 Guidelines checklist, a document that aims to improve experimental reporting and reproducibility of animal studies for purposes of post-publication data analysis and reproducibility: https://arriveguidelines.org/sites/arrive/files/documents/Author%20Checklist%20-%20Full.pdf Please include your completed checklist as a Supporting Information file. Note that if your paper is accepted for publication, this checklist will be published as part of your article.

Reviewers' comments:

Reviewer's Responses to Questions

**Comments to the Author**

1. Is the manuscript technically sound, and do the data support the conclusions?

Reviewer #1: Yes

Reviewer #2: Yes

2. Has the statistical analysis been performed appropriately and rigorously? 

Reviewer #1: Yes

Reviewer #2: I Don't Know

3. Have the authors made all data underlying the findings in their manuscript fully available?

Reviewer #1: Yes

Reviewer #2: Yes

4. Is the manuscript presented in an intelligible fashion and written in standard English?

Reviewer #1: Yes

Reviewer #2: Yes

5. Review Comments to the Author

Reviewer #1: Thank you for your work and your report on these 2 methods for cecal decompression in horses. Largely this study is well-conducted and the paper is well written. My biggest concern is the low number of subjects in a healthy population of horses that may make it challenging to broadly extrapolate to a larger population of sick horses with cecal distension. However, this is a great start and will serve as a foundation to more clinical studies. A few specific comments by line number follow:

• Line 42: Peritonitis, even if not clinically detectable, could arguably still be considered a complication. If no horses had complications, then there should really be no difference between method of cecal decompression. I would specify that there were no clinically detectable complications. Or in line 455 you say 'no major complications' - I think that is appropriate.

• Line 46: Please check, is there a scientific reason to not call this a 'fever'?

• Line 81: 'causes more tissue damage'. It also causes a bigger hole for leakage.

• Line 114: Here you say 'large intestinal trocarization' site. In other places (ex. line 120) you say 'cecal trocarization'. While both are correct, cecal tocarization is more precise and I would suggest you stay consistent throughout the manuscript.

• Line 122: I would not say that 'and in which serial blood and peritoneal fluid samples were collected' here and when describing group 2 and the control group. Since sample collection was done the same for all groups, it would make it easier to read if you merely say this after describing all three groups. The reader will be able to focus on the differences between the groups.

• Line 127: As horses were enrolled in the study, they were randomly assigned to one of the three treatment groups. I think this is a random crossover design. If so, please specify.

• Line 136: Previously you used hyperthermia instead of 'pyrexia'. Please stay consistent. I would suggest 'pyrexia' or 'fever' unless you think that the increased temperature could be due to something else like increased ambient temperature.

• Line 150: Should be 'a number 15 blade'. Not 'gauge'.

• LIne 237: Best word choice? Hyperthermia, Pyrexia, or Fever?

• Line 460-461: antimicrobials do not reduce abdominal contamination, they merely help to combat infection that results.

• Discussion: I think it is also important to mention that these were healthy horses without cecal distension. Please comment in the discussion how the authors feel that horses with cecal tympany may be different which is the population of horses that this procedure would be performed on routinely. I would expect more leakage from around the cannula/needle since the gas is under more pressure.

• Figure 1: The image on my PDF copy of the manuscript is blurry and it is difficult to see details. Hopefully this will be rectified in the final published copy.

• Figure 2: I feel that this figure is nice to give a visual representation of how the peritoneal fluid changed over time. However the labels on the tubes are difficult to read in the photograph. I would recommend that text labels be added to the image over the respective tubes to indicate the time points at which these samples were drawn. (ex. T0, T2, T3, etc).

• Perhaps consider adding some graphs (in addition to the tables) to show how the data compared for the three groups for some of the variables.

Reviewer #2: The paper describes the potential for complications following transabdominal cecal trocharization of 6 healthy horses using a cecal needle or a cattle trocar. While only 6 horses were used the cross over study design with random assignment three month period of washout of control, cecal needle or cattle trocar appears suitable for interpretation of finding, and is in keeping with the principles of the 3 R’s. Horses were monitored for signed of clinically relevant peritonitis (CRP) and for a wide range of laboratory analyses both on blood and peritoneal fluid for clinical pathologic changes reflective of peritonitis.

The study was clearly written and most procedure well described, with very good transparency of the results. The finding that both methods of cecal trocarization in normal healthy horses led to some degree of peritonitis based on laboratory but not CRP appear well substantiated. In most places within the manuscript please write “a needle trocar “ or “a cattle trocar”.

The authors refer to many key scientific papers and hypothesis based studies that are highly relevant for this work. However, there are many references included that are largely book chapters or non peer reviewed works ( refs 5,7,8,9,11,12,20,21,22 among others) that do not appear essential to this work.

My further comments and general critique are as below.

Firstly, the authors describe the monitoring for CRP, with inclusion of use of both objective (heart rate (HR), respiratory rate (RR) temperature (ºC), and subjective (Pain assessment, general appearance, auscultation of gastrointestinal motility, and perfusion indices (mucous

membrane color and capillary refill)). Was there any scoring system used for these subjective indices? In particular, how were pain, general appearance, intestinal motility assessed? For example, use of pain face in horses is now well accepted as a semiquantitative measure (see An equine pain face. Gleerup KB, Forkman B, Lindegaard C, Andersen PH. Vet Anaesth Analg. 2015 Jan;42(1):103-14. doi: 10.1111/vaa.12212. Epub 2014 Jul 31). Similarly, what criteria were used for ”general appearance” and can the authors clarify how the assessment of degree of intestinal motility was graded for comparison?

These aspects should be more transparent, as the authors are suggesting that CRP did not occur in any of the horses, yet in the manuscript’s current form the reader must simply accept this conclusion despite incomplete evidence.

The statistics used for this paper appear suitable but should be reviewed by another expert as I lack competence to make further comment. However presentation of confidence intervals would have been preferable to the SD used, as for example Table 8, the data as presented would suggest that for G2 ceruloplasmin was already higher at t=0 in comparison to CG. Please comment.

Secondly, it would be importance within the discussion to address the aspect of the findings being relevant in normal healthy horses, and if they can be applicable to horses with colic that are deemed in need of cecal trocarization. In such cases, please also comment on the relevance of this study’s trocar size difference, and whether s gas removal from a severely distended cecum may be a factor to weigh in on the finding in this work.

Other minor comments are as follows.

Line 215 … blood cell count (< 5.5 x 103.. should be 103

Line 181 “Polyacrylamide Gel Electrophoresis” This section is heavy on laboratory analysis, but I would ask the authors what clinical significance it has for this study?

Line 182 …were centrifuged at 3,000 rpm … please convert to x g

Line 434. “It is known that cecal puncture promotes a solution of continuity with the

abdominal cavity and represents a peritoneal insult”. Please reword, I am unsure what is meant by this.

Line 439 “As demonstrated in this study, intense migration of neutrophils into the peritoneal cavity occurs due to the local release of chemotactic substances [32], which act in the primary cellular defense against microorganisms and have intense phagocytic activity [29]. “

As written this suggest that the work included measurement of chemotactic factors. Should rewrite to clarify only what the authors studied : Ie Migration of neutrophils into the peritoneal cavity as found in this study occurs due to the local release of chemotactic substances [32], which act in the primary cellular defense against microorganisms and have intense phagocytic activity [29].

Line 442: “Moreover, the increase of alkaline phosphatase enzyme in peritoneal fluid observed in this study is related to inflammatory processes and the presence of intestinal bacteria in the fluid [33,34]”

As above, the authors did not assess for bacterial presence, and simply inflammation can have increase release of alkaline phosphatase,

Suggest rephrasing as “Moreover, the increase of alkaline phosphatase enzyme in peritoneal fluid observed in this study was likely related to induction of an inflammatory processes due to trocarization and potentially the presence of intestinal bacteria in the fluid

Lines 447-447 “Therefore, the results of this study show that serial abdominocentesis is not responsible for the increase in total proteins, total nucleated cells, and alkaline phosphatase in the peritoneal fluid.

These particular findings are highly relevant clinically, and the authors deserve to highlight the finding more that just this one sentence.

Line 471 “14 days of evaluation could be justified by the increase”… , rewrite to “14 days of evaluation could be explained by the increase…

Line 478 …“once the horses.. change to …“ since the horses…”

6. PLOS authors have the option to publish the peer review history of their article (what does this mean?). If published, this will include your full peer review and any attached files.

Reviewer #1: **Yes: **Andy Niehaus

Reviewer #2: **Yes: **John Pringle DVM, PhD

---

## [Author Response · Author response to Decision Letter 0]

5 Oct 2022

PONE-D-22-22467

Complications following transcutaneous cecal trocarization in horses with cattle trocar and cecal needle

PLOS ONE

Reviewer #1: 

Thank you for your work and your report on these 2 methods for cecal decompression in horses. Largely this study is well-conducted and the paper is well written. My biggest concern is the low number of subjects in a healthy population of horses that may make it challenging to broadly extrapolate to a larger population of sick horses with cecal distension. However, this is a great start and will serve as a foundation to more clinical studies. 

Answer: We appreciate the reviewer's 1 comments, and we do believe this will be a high impact study to be cited.

A few specific comments by line number follow:

• Line 42: Peritonitis, even if not clinically detectable, could arguably still be considered a complication. If no horses had complications, then there should really be no difference between method of cecal decompression. I would specify that there were no clinically detectable complications. Or in line 455 you say 'no major complications' - I think that is appropriate.

Answer: Suggestion accepted.

• Line 46: Please check, is there a scientific reason to not call this a 'fever'?

Answer: There is no reason not to call this a fever, we've changed the sentence.

• Line 81: 'causes more tissue damage'. It also causes a bigger hole for leakage.

Answer: We have changed the text.

• Line 114: Here you say 'large intestinal trocarization' site. In other places (ex. line 120) you say 'cecal trocarization'. While both are correct, cecal tocarization is more precise and I would suggest you stay consistent throughout the manuscript.

Answer: We have changed the term “large intestine trocarization” to “cecal trocarization” throughout the manuscript when referring to our objectives, methods, results and conclusions, since we performed cecal trocarization. 

• Line 122: I would not say that 'and in which serial blood and peritoneal fluid samples were collected' here and when describing group 2 and the control group. Since sample collection was done the same for all groups, it would make it easier to read if you merely say this after describing all three groups. The reader will be able to focus on the differences between the groups.

Answer: Suggestion accepted. The paragraph was rephrased. 

• Line 127: As horses were enrolled in the study, they were randomly assigned to one of the three treatment groups. I think this is a random crossover design. If so, please specify.

Answer: Suggestion accepted.

• Line 136: Previously you used hyperthermia instead of 'pyrexia'. Please stay consistent. I would suggest 'pyrexia' or 'fever' unless you think that the increased temperature could be due to something else like increased ambient temperature.

Answer: The increase in temperature is not due to the increase in ambient temperature. We changed the term throughout the manuscript and stayed consistent (fever).

• Line 150: Should be 'a number 15 blade'. Not 'gauge'.

Answer: The error has been fixed.

• Line 237: Best word choice? Hyperthermia, Pyrexia, or Fever?

Answer: We have replaced all terms about increased temperature in the text with "fever".

• Line 460-461: antimicrobials do not reduce abdominal contamination, they merely help to combat infection that results.

Answer: This sentence was removed from the text.

• Discussion: I think it is also important to mention that these were healthy horses without cecal distension. Please comment in the discussion how the authors feel that horses with cecal tympany may be different which is the population of horses that this procedure would be performed on routinely. I would expect more leakage from around the cannula/needle since the gas is under more pressure.

Answer: We added the following sentence to the manuscript’s main limitations:

“It is also important to note that these were healthy horses, without a severely distended cecum. In horses with colic, where trocarization of the cecum is considered necessary, would probably have more leakage from around the trocar/needle following puncture since the cecum is under more pressure.”

• Figure 1: The image on my PDF copy of the manuscript is blurry and it is difficult to see details. Hopefully this will be rectified in the final published copy.

Answer: Possibly it is a PDF copy problem. The original image is not blurry.

• Figure 2: I feel that this figure is nice to give a visual representation of how the peritoneal fluid changed over time. However the labels on the tubes are difficult to read in the photograph. I would recommend that text labels be added to the image over the respective tubes to indicate the time points at which these samples were drawn. (ex. T0, T2, T3, etc).

Answer: Suggestion accepted.

• Perhaps consider adding some graphs (in addition to the tables) to show how the data compared for the three groups for some of the variables.

Answer: Suggestion accepted.

Reviewer #2: 

The paper describes the potential for complications following transabdominal cecal trocharization of 6 healthy horses using a cecal needle or a cattle trocar. While only 6 horses were used the cross over study design with random assignment three month period of washout of control, cecal needle or cattle trocar appears suitable for interpretation of finding, and is in keeping with the principles of the 3 R’s. Horses were monitored for signed of clinically relevant peritonitis (CRP) and for a wide range of laboratory analyses both on blood and peritoneal fluid for clinical pathologic changes reflective of peritonitis.

The study was clearly written and most procedure well described, with very good transparency of the results. The finding that both methods of cecal trocarization in normal healthy horses led to some degree of peritonitis based on laboratory but not CRP appear well substantiated. In most places within the manuscript please write “a needle trocar “ or “a cattle trocar”.

The authors refer to many key scientific papers and hypothesis based studies that are highly relevant for this work. However, there are many references included that are largely book chapters or non peer reviewed works ( refs 5,7,8,9,11,12,20,21,22 among others) that do not appear essential to this work.

Answer: We appreciate the reviewer's 2 comments, and we have accepted all the suggestions aiming to enhance the manuscript quality.

My further comments and general critique are as below.

Firstly, the authors describe the monitoring for CRP, with inclusion of use of both objective (heart rate (HR), respiratory rate (RR) temperature (ºC), and subjective (Pain assessment, general appearance, auscultation of gastrointestinal motility, and perfusion indices (mucous membrane color and capillary refill)). Was there any scoring system used for these subjective indices? In particular, how were pain, general appearance, intestinal motility assessed? For example, use of pain face in horses is now well accepted as a semiquantitative measure (see An equine pain face. Gleerup KB, Forkman B, Lindegaard C, Andersen PH. Vet Anaesth Analg. 2015 Jan;42(1):103-14. doi: 10.1111/vaa.12212. Epub 2014 Jul 31). Similarly, what criteria were used for ”general appearance” and can the authors clarify how the assessment of degree of intestinal motility was graded for comparison? These aspects should be more transparent, as the authors are suggesting that CRP did not occur in any of the horses, yet in the manuscript’s current form the reader must simply accept this conclusion despite incomplete evidence.

Answer: Qualitative analyses were performed using a 0-to-3 scoring system. The paragraph was rephrased. 

“Heart rate (HR), respiratory rate (RR), temperature (ºC), and perfusion indices (mucous membrane color and capillary refill) were evaluated. Auscultation of digestive sounds (bowel movements) and behavior (appearance, appetite, sweating, kicking at abdomen, and pawing on the floor) were performed with a general score ranging from 0 to 3 (19). These parameters were monitored prior to starting the cecal trocarization (T0) and 2 (T2), 6 (T6) and 12 (T12) hours after the first sampling and every 12 hours during 14 days after the procedure.”

The statistics used for this paper appear suitable but should be reviewed by another expert as I lack competence to make further comment. However presentation of confidence intervals would have been preferable to the SD used, as for example Table 8, the data as presented would suggest that for G2 ceruloplasmin was already higher at t=0 in comparison to CG. Please comment.

Answer: The data were analyzed by a statistician from the University of São Paulo and the decision of presentation of standard deviation (SD) was made since we have used a small sample size, representing a greater individual variability. We thought that the data as presented might be more transparent to the reader. 

Regarding ceruloplasmin, we have accepted the reviewer suggestion and the text was modified (there were no differences between groups at the same time).

Secondly, it would be importance within the discussion to address the aspect of the findings being relevant in normal healthy horses, and if they can be applicable to horses with colic that are deemed in need of cecal trocarization. In such cases, please also comment on the relevance of this study’s trocar size difference, and whether s gas removal from a severely distended cecum may be a factor to weigh in on the finding in this work.

Answer: The difference between normal healthy horses and those with colic and tympanic cecum was discussed in the limitations of the study, as also suggested by the reviewer 1. This study shows encouraging results for future clinical trials, which would potentially show similar results. We would expect more impacting systemic alterations because of the acute abdomen general systemic alterations. 

“It is also important to note that these were healthy horses, without a severely distended cecum. In horses with colic, where trocarization of the cecum is considered necessary, would potentially have more leakage from around the trocar/needle following puncture since the cecum is under more pressure.”

Other minor comments are as follows.

• Line 215 … blood cell count (< 5.5 x 103.. should be 103)

Answer: Suggestion accepted.

• Line 181 “Polyacrylamide Gel Electrophoresis” This section is heavy on laboratory analysis, but I would ask the authors what clinical significance it has for this study?

Answer: Acute-phase proteins analysis can be performed by different techniques, and there are specific methodological differences among the available methods. Electrophoresis of horse serum using a wide range of support media and techniques has been reported. For this reason, we have chosen to describe it in more detail, as this will allow reproducibility of the study and reliable comparison with similar researches and methodologies. 

• Line 182 …were centrifuged at 3,000 rpm … please convert to x g

Answer: Suggestion accepted. Converted to 1,161 g.

• Line 434. “It is known that cecal puncture promotes a solution of continuity with the

abdominal cavity and represents a peritoneal insult”. Please reword, I am unsure what is meant by this.

Answer: Suggestion accepted. “It is known that cecal puncture may promote some degree of leakage of cecal content into the peritoneal cavity and represents a peritoneal insult [24].”

• Line 439 “As demonstrated in this study, intense migration of neutrophils into the peritoneal cavity occurs due to the local release of chemotactic substances [32], which act in the primary cellular defense against microorganisms and have intense phagocytic activity [29].” As written this suggest that the work included measurement of chemotactic factors. Should rewrite to clarify only what the authors studied : Ie Migration of neutrophils into the peritoneal cavity as found in this study occurs due to the local release of chemotactic substances [32], which act in the primary cellular defense against microorganisms and have intense phagocytic activity [29].

Answer: Suggestion accepted.

• Line 442: “Moreover, the increase of alkaline phosphatase enzyme in peritoneal fluid observed in this study is related to inflammatory processes and the presence of intestinal bacteria in the fluid [33,34]”.

As above, the authors did not assess for bacterial presence, and simply inflammation can have increase release of alkaline phosphatase,

Suggest rephrasing as “Moreover, the increase of alkaline phosphatase enzyme in peritoneal fluid observed in this study was likely related to induction of an inflammatory processes due to trocarization and potentially the presence of intestinal bacteria in the fluid

Answer: Suggestion accepted.

• Lines 447-447 “Therefore, the results of this study show that serial abdominocentesis is not responsible for the increase in total proteins, total nucleated cells, and alkaline phosphatase in the peritoneal fluid.

These particular findings are highly relevant clinically, and the authors deserve to highlight the finding more that just this one sentence.

Answer: Suggestion accepted. Another sentence was inserted into discussion section (lines 433-434): “Serial abdominocentesis was not responsible for hematological or peritoneal fluid alterations, as evidenced in CG.”

• Line 471 “14 days of evaluation could be justified by the increase”… , rewrite to “14 days of evaluation could be explained by the increase…

Answer: Suggestion accepted.

• Line 478 …“once the horses.. change to …“ since the horses…”

Answer: Suggestion accepted.

---

## [Decision Letter · Decision Letter 1]

28 Oct 2022

Complications following transcutaneous cecal trocarization in horses with a cattle trocar and a cecal needle

PONE-D-22-22467R1

Dear Dr. Dória,

We’re pleased to inform you that your manuscript has been judged scientifically suitable for publication and will be formally accepted for publication once it meets all outstanding technical requirements.

Kind regards,

Benito Soto-Blanco, DVM, MSc, PhD

Academic Editor

PLOS ONE

Reviewers' comments:

Reviewer's Responses to Questions

**Comments to the Author**

1. If the authors have adequately addressed your comments raised in a previous round of review and you feel that this manuscript is now acceptable for publication, you may indicate that here to bypass the “Comments to the Author” section, enter your conflict of interest statement in the “Confidential to Editor” section, and submit your "Accept" recommendation.

Reviewer #1: All comments have been addressed

Reviewer #2: All comments have been addressed

2. Is the manuscript technically sound, and do the data support the conclusions?

Reviewer #1: Yes

Reviewer #2: Yes

3. Has the statistical analysis been performed appropriately and rigorously? 

Reviewer #1: Yes

Reviewer #2: Yes

4. Have the authors made all data underlying the findings in their manuscript fully available?

Reviewer #1: Yes

Reviewer #2: Yes

5. Is the manuscript presented in an intelligible fashion and written in standard English?

Reviewer #1: Yes

Reviewer #2: Yes

6. Review Comments to the Author

Reviewer #1: Thank you for this manuscript. The authors have addressed my previous concerns with this manuscript. I feel that this is a nice study that will guide clinicians in making appropriate decisions regarding cecal trocarization in horses.

Reviewer #2: Thank you for the response and revisions based on my and Reviewer #1 comments. My only query that remains is the ceruloplasmin where the data appear to suggest t=0 being markedly lower in the control group, much higher at the start for G1 and G2- while these still were not significantly higher at T= 24 and T=72 than their base lines. Thus I expect the reader to also question whether this "rise over time" -as found statistically- may have any underlying biologic relevance.

Perhaps a comment within the discussion would be appropriate regarding the relative strength of evidence of these changes in relation to the many others (far more convincing- such as nucleated cells, neutrophils.. ).

7. PLOS authors have the option to publish the peer review history of their article (what does this mean?). If published, this will include your full peer review and any attached files.

Reviewer #1: **Yes: **Andy Niehaus

Reviewer #2: **Yes: **John Pringle

---

## [Editor Report · Acceptance letter]

14 Nov 2022

PONE-D-22-22467R1 

Complications following transcutaneous cecal trocarization in horses with a cattle trocar and a cecal needle 

Dear Dr. Dória:

I'm pleased to inform you that your manuscript has been deemed suitable for publication in PLOS ONE. Congratulations! Your manuscript is now with our production department. 

Kind regards, 

on behalf of

Dr. Benito Soto-Blanco 

Academic Editor

PLOS ONE